



# High-precision 1′×1′ bathymetric model of Philippine Sea inversed from marine gravity anomalies

Dechao An[1,2], Jinyun Guo[1*], Xiaotao Chang[3], Zhenming Wang[3], Yongjun Jia[4], Xin Liu[1], Valery Bondur[5], Heping Sun[6]

5  [1]College of Geodesy and Geomatics, Shandong University of Science and Technology, Qingdao 266590, China
[2]School of Geospatial Engineering and Science, Sun Yat-Sen University, Zhuhai 519082, China.
[3]Land Satellite Remote Sensing Application Center, Ministry of Natural Resources, Beijing 100048, China
[4]National Satellite Ocean Application Service, Ministry of Natural Resources, Beijing 100081, China
[5]AEROCOSMOS Research Institute for Aerospace Monitoring, Moscow 105064, Russia
10  [6]State Key Laboratory of Geodesy and Earth's Dynamics, Innovation Academy for Precision Measurement Science and Technology, Chinese Academy of Sciences, Wuhan 430071, China

Corresponding author: Jinyun Guo (jinyunguo1@126.com)

**Abstract.** The Philippine Sea, located at the edge of the Northwest Pacific Ocean, possesses complex seabed topography. Developing a high-precision bathymetric model for this region is of paramount importance as it provides fundamental geoinformation essential for Earth observation and marine scientific research, including plate motion, ocean circulation, and hydrological characteristics. The gravity-geologic method (GGM), based on marine gravity anomalies, serves as an effective bathymetric prediction technique. To further strengthen the prediction accuracy of conventional GGM, we introduce the improved GGM (IGGM). The IGGM considers the effects of regional seafloor topography by employing weighted averaging to more accurately estimate the short-wavelength gravity component, along with refining the subsequent modeling of long-wavelength gravity component. In this paper, we focus on seafloor topography modeling in the Philippine Sea based on the IGGM, combining shipborne bathymetric data with the SIO V32.1 gravity anomaly. To reduce computational complexity, the optimal parameter values required for IGGM are first calculated before the overall regional calculation, and then, based on the terrain characteristics and distribution of sounding data, we selected four representative local sea areas as the research objects to construct the corresponding bathymetric models using GGM and IGGM. The analysis indicates that the precisions of the IGGM models in four regions are improved to varying degrees, and the optimal calculation radius is 2′. Based on the above finding, a high-precision 1'×1' bathymetric model of the Philippine Sea (5° N–35° N, 120° E–150° E), known as the BAT_PS model, is constructed using IGGM. Results demonstrate that the BAT_PS model exhibits a higher overall precision compared to GEBCO, topo_25.1, and DTU18 models at single-beam shipborne bathymetric points.



**Bathymetric model of Philippine Sea ( BAT_PS model) inversed from marine gravity anomalies**



## 1 Introduction

The Philippine Sea serves as a convergence zone for continental and oceanic plates, resulting in frequent and intense plate tectonic activity (Richter and Ali, 2015; Lallemand, 2016; Holt et al., 2018). The Philippine Sea has complex topography, including trenches, island arcs, ridges, seamounts, basins, and rifts, exhibiting typical characteristics of a trench-arc-basin

system. Additionally, it stands among the highly dynamic areas for geospatial exploration globally, attracting significant attention as a focal area for international scientific research. As fundamental geoinformation for marine scientific research, the accurate acquisition and application of high-precision seafloor topography are crucial (Kunze et al., 2004; Jena et al., 2012; Hirt and Rexer, 2015; Hu et al., 2020). They provide necessary information to guarantee an in-depth comprehension of the marine environment and support marine development and governance (Ryabinin et al., 2019; Wolfl et al., 2019).

The Seabed 2030 project, a collaboration under the General Bathymetric Chart of the Oceans (GEBCO) and the Nippon Foundation, strives to achieve the creation of a comprehensive global mapping of the seafloor topography by 2030 (Mayer et al., 2018). In May 2023, the International Hydrographic Organization (IHO) announced that Seabed 2030 project had collected mapping data covering 24.9% of the seabed. Additionally, multiple GEBCO models have been released, which integrate single-beam shipborne bathymetric data, high-resolution multi-beam shipborne bathymetric data, and predicted

bathymetry based on satellite altimetry data. For unexplored areas that have not been covered by bathymetric data, the marine gravity field derived from satellite altimetry data (Zhu et al., 2022; Zhou et al., 2023; Wei et al., 2023) is currently the main source for constructing high-precision and high-resolution seafloor bathymetric models, as compared to other observation methods (Bondur and Grebenyuk, 2001; Smith, 2004; Hilldale and Raff, 2008; Yeu et al., 2018; Tozer et al., 2019; Yu et al., 2022; Xu et al., 2023; Yu et al., 2023).

The prediction of seafloor topography based on satellite altimetry data has undergone a progression from one–dimensional to two–dimensional approaches, and various methods have been developed, such as the gravity-geology method (GGM), the S&S method (Smith and Sandwell, 1994), the frequency-domain method (Parker, 1973; Watts, 1978), and the simulated annealing method (Yang et al., 2018, 2020). The GGM, proposed by Ibrahim (1972), was initially used to measure the height of bedrock beneath sediments in land areas, and subsequently employed in the inversion studies of seabed topography. Based

on gravity anomalies derived from satellite altimetry missions, Smith and Sandwell (1994) constructed the corresponding bathymetric model and observed a significant correlation between seafloor topography and gravity anomaly within the wavelength band of 15 to 160 km. Yang et al. (2018) predicted seafloor topography in the Western Pacific Ocean using vertical gravity gradient data through simulated annealing. Compared with the shipborne measured depth, the root mean square of the prediction result was 236 m, representing a 22 % improvement in accuracy over SIO bathymetric model.

However, the simulated annealing in the spatial domain includes forward and inverse modeling, requiring significant computational resources for modifying the initial model through continuous iteration. While the GGM and S&S method use the linear relationship between seafloor topography and gravity data to construct empirical functions, the accuracy in regions with complex topography needs improvement. The frequency-domain method is based on the spectral relation between



seafloor topography and marine gravity field, using the first-order term of the Parker formula to approximate bathymetry.

This method omits the effect of higher-order terms and includes several complex geophysical parameters.

As one of the commonly used methods for bathymetric prediction, GGM can use shipborne bathymetry and gravity anomalies to build a bathymetric model with high accuracy (Nagarajan, 1994; Wei et al., 2021; Kim et al., 2010; Hsiao et al., 2011; Annan and Wan 2020). The GGM divides the gravity anomaly into short–wavelength component and long-wavelength component. The short-wavelength gravity can be calculated using the Bouguer plate formula, while the estimation of the

long-wavelength gravity is particularly important. Annan and Wan (2020) adopted an adaptive mesh form to approximate the long-wavelength gravity with a prediction accuracy of 180.20 m in the Gulf of Guinea, comparable to the accuracy of the ETOPO1 model and SIO model. In the conventional GGM, the process of calculating short-wavelength gravity from sea depth according to the Bouguer plate formula is a one-to-one mapping, which ignores the gravity effect caused by surrounding seafloor topography. To overcome this limitation, An et al. (2022) proposed the improved GGM (IGGM)

considering the effect of regional seafloor topography, which introduced weighting parameters into the Bouguer slab formula using the depth at the control point and the distance between surrounding points and control points to evaluate the short-wavelength gravity effect from the seafloor topography around the control point. The IGGM finely calculates the short-wavelength gravity by weighted averaging and refines the subsequent long-wavelength gravity modeling, significantly enhancing the overall accuracy of the bathymetric model.

Due to its prominent strategic position and complex marine environment, constructing an accurate bathymetric model of the Philippine Sea is of utmost importance. Therefore, this paper focuses on constructing a higher-precision bathymetric model for this region, utilizing the IGGM in combination with shipborne bathymetric data and the SIO V32.1 gravity anomaly. To effectively enhance the calculation efficiency of the bathymetric model, and be guided by the topographic characteristics and the distribution of shipborne bathymetric data, four representative areas are chosen as the research objects before the overall

solution. This preliminary stage involves exploring the optimal values of required parameters and assessing the applicability of IGGM for the Philippine Sea. Subsequently, a 1'×1' bathymetric model of the Philippine Sea (5° N–35° N, 120° E–150° E) is obtained using priori values of certain parameters. Finally, the accuracy of the bathymetric model is evaluated by comparing with existing models, including the GEBCO_2022, topo_25.1, and DTU18 models.

## 2 Data Sources

The Philippine Sea, located between the two island chains in the Western Pacific Ocean, consists of Philippine Basin, Parece Vela Basin, Shikoku Basin, Kyushu-Palau Ridge, Izu-Ogasawara Trench, Mariana Trenches and Ridge. Recognizing the importance of the seafloor topography as essential geoinformation within this region, this paper focuses on the Philippine Sea (5° N–35° N, 120° E–150° E) as the study area, which presents significant challenges for shipborne bathymetry and the inversion of seafloor topography based on satellite altimetry data due to its complex terrain and geomorphology.



This paper used single-beam shipborne bathymetry obtained from the National Centers for Environmental Information (NCEI). However, due to the large time span over which bathymetric data were acquired, some data collected before GPS became operational were often poorly localized and contained significant measurement errors (Smith, 1993). Therefore, it is necessary to use a bathymetric model with higher accuracy as a reference (e.g., GEBCO_2022 model) for ensuring the quality of the shipboard bathymetric data. Comparing the shipborne bathymetric data with the reference model, the gross

errors exceeding "$3\sigma$" were eliminated from the original bathymetric data. The gravity anomalies were obtained from the SIO V32.1 model with a resolution of 1 arcmin, which was released in August 2022. The V32.1 model in the study area is shown in Fig. 1.

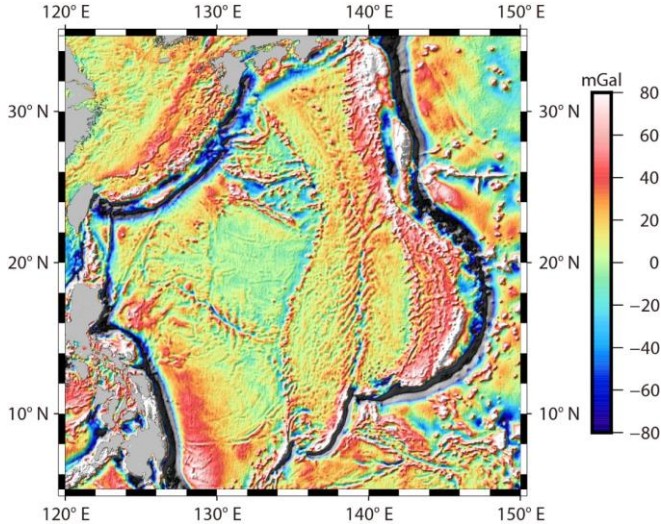

**Figure 1.** SIO V32.1 gravity anomaly model in the study area.

The GEBCO_2022 Grid, released in June 2022, is a continuous, global terrain model with a grid spacing of 15 arcsec (GEBCO, 2022), briefly referred to as the GEBCO model. Over time, GEBCO develops a range of bathymetric datasets and products. GEBCO provides a Type Identifier grid that specifies the type of source data used for each grid cell, including single-beam shipborne bathymetry, multi-beam shipborne bathymetry, LiDAR bathymetry, prediction depths derived from satellite gravity, and grid interpolation depths, etc. By continually collecting and incorporating the latest and most relevant

bathymetric data, the GEBCO series of models represents the cutting-edge level of accuracy for terrain models in the world. Furthermore, the DTU18 and topo_25.1 bathymetric models were used in this paper, both of which have a grid resolution of 1 arcmin. The DTU18 model was released by the Technical University of Denmark (DTU) in 2019. SIO has been continuously updating its bathymetric models for an extended period, and the topo_25.1 model, released in January 2023, is the latest version of the current series.





## 3 Methodology

### 3.1 Principle of the improved gravity-geologic method

In fact, there is a nonlinear relationship between seafloor topography and marine gravity anomalies. In many geodetic calculations, the nonlinear issue can be linearized by employing a suitable reference field (Hwang, 1999). The principle is to decompose the gravity anomaly into two main components: the regional gravity field representing the long-wavelength gravity and the residual gravity field representing the short-wavelength gravity. Among them, the long-wavelength component is generated by deeper mass variations and the short-wavelength component is derived from variations of local bedrock under sediment (Kim et al., 2010; Hsiao et al., 2016). The gravity anomaly ( $\Delta g$ ) is composed of the long-wavelength component ( $\Delta g_{reg}$ ) and the short-wavelength component ( $\Delta g_{res}$ ):

$$\Delta g = \Delta g_{reg} + \Delta g_{res} . \tag{1}$$

Using the known depth at control points to obtain the short-wavelength gravity component ( $\Delta g_{res}^{j_n}$ ):

$$\Delta g_{res}^{j_n} = 2\pi G \Delta\rho(E_{j_n} - D) , \tag{2}$$

where $G$ is the gravitational constant ($6.672\times10^{-8}$ cm$^3$/g·s$^2$); $\Delta\rho$ is the density contrast (g/cm$^3$) between seawater and bedrock; $E_{j_n}$ is the depth at $j_n$ point; $D$ is a reference datum, which is generally the deepest depth of control points.

The GGM uses the Bouguer slab Eq. (2) to compute the short-wavelength component of the sea surface point, which represents a single-point calculation form as shown in the left part of Fig. 2 and is not considered rigorous. In response to this problem, IGGM refines the long-wavelength gravity model by considering the short-wavelength gravity effect of regional seafloor topography as shown in the right part of Fig. 2. Using the control point ( $j_n$ ) as the center, $R$ is the calculated radius for estimating the effect of seafloor topography, $m$ is the number of sounding points within the calculation range, and $j_n^m$ is the encompassing shipborne sounding point.





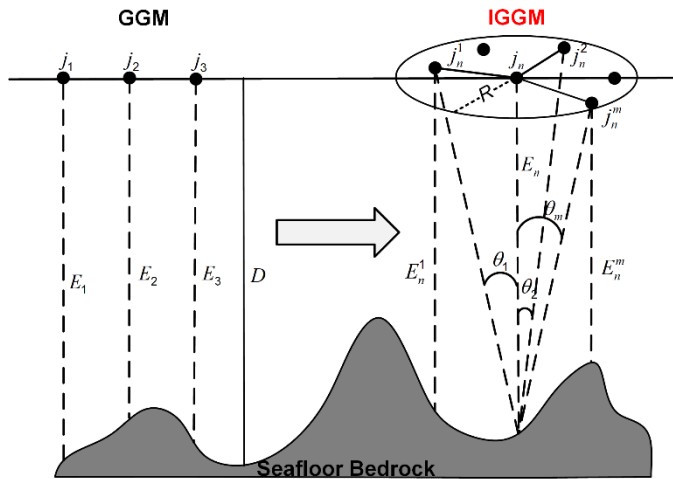

**Figure 2.** Schematic geometries of GGM and IGGM.

Based on Eq. (2), introducing $\cos^k \theta_m$ as the weight parameter, the short-wavelength gravity effect ($\Delta g_{res}^{j_n^m}$) of the surrounding shipborne points on the control point is:

$$\Delta g_{res}^{j_n^m} = 2\pi G \Delta\rho(\cos^k \theta_m)(E_{j_n^m} - D) \,, \tag{3}$$

Then, the short-wavelength component ($\Delta g_{res}^{j_n}$) is corrected using weighted averaging:

$$\begin{aligned}\Delta g_{res}^{j_n} &= 2\pi G \Delta\rho \frac{(\cos^k \theta_1)(E_{j_n^1} - D) + (\cos^k \theta_2)(E_{j_n^2} - D) + \cdots + (\cos^k \theta_m)(E_{j_n^m} - D)}{\cos^k \theta_1 + \cos^k \theta_2 + \cdots + \cos^k \theta_m} \\ &= 2\pi G \Delta\rho \frac{\displaystyle\sum_{s=1}^{m}(\cos^k \theta_s)(E_{j_n^s} - D)}{\displaystyle\sum_{s=1}^{m}(\cos^k \theta_s)}\end{aligned} \,. \tag{4}$$

Subtracting the refined short-wavelength component calculated by Eq. (4) from the gravity anomaly to obtain the long-wavelength component ($\Delta g_{reg}^{j_n}$) at the control point $j_n$:

$$\Delta g_{reg}^{j_n} = \Delta g^{j_n} - \Delta g_{short}^{j_n} \,. \tag{5}$$

The long-wavelength component at control points is gridded using a tension spline function to obtain the long-wavelength gravity field. The long-wavelength component ($\Delta g_{reg}^{i}$) at any point $i$ can be calculated through cubic spline interpolation. Subsequently, the short-wavelength component ($\Delta g_{res}^{i}$) at the prediction point $i$ is:



$$\Delta g^i_{res} = \Delta g^i - \Delta g^i_{reg} \ . \tag{6}$$

Finally, based on a variation of the Bouguer formula, the predicted depth ($E_i$) at point $i$ is inversely calculated using the short-wavelength component:

$$E_i = \frac{\Delta g^i_{res}}{2\pi G\Delta\rho} + D \ . \tag{7}$$

**3.2 Calculation process of improved gravity-geologic method**

The key of IGGM is how to determine and calculate the optimal value of $\Delta\rho$, $k$ and calculation radius $R$, so that gravity anomalies can better characterize the basic geoinformation of seafloor topography. In this study, referring to the process of determining $\Delta\rho$ in GGM, the parameters $\Delta\rho$, $R$ and $k$ are also calculated by iterative method in IGGM. The value ranges of $\Delta\rho$, $R$ and $k$ are set as 0.1 g/cm³–5 g/cm³, 0–30', and 0.1–15, and their corresponding iteration steps are 0.1 g/cm³, 0.1', and 0.1, respectively. The correlation coefficient and STD between the predicted depth and the shipborne bathymetry at points $i$ are analyzed under the conditions of different parameter values. When the correlation coefficient is the largest and the STD error is the smallest between predicted depths and measured depths (i.e. $\max_{1\le i\le n} CC_i$ and $\min_{1\le i\le n} STD_i$), the corresponding parameters are the optimal values. In the iterative process of IGGM, the parameters are independent of each other. The optimal values of $\Delta\rho$, $R$ and $k$ are determined after all the iterations of the parameters have been completed. Fig. 3 presents the flowchart of IGGM, providing a detailed visual representation of the process.

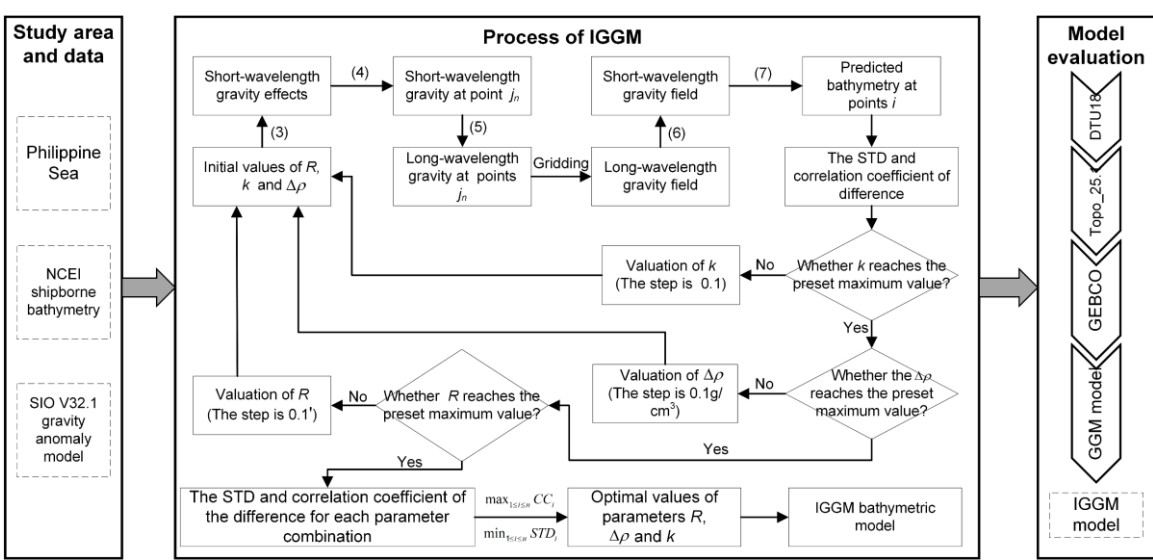

**Figure 3.** Flowchart of the IGGM.



## 4 Results and discussion

### 4.1 Parameter determination and applicability verification of IGGM

Due to the large size of the Philippine Sea and the substantial amount of bathymetric data, improving the calculation efficiency of the bathymetric model becomes essential. If certain unknown parameters can be determined before the overall model calculation, it can effectively achieve the purpose. In view of this, according to the different topographic features and bathymetric data distribution, four representative local areas within the Philippine Sea were first selected as the research objects, analysing and determining the unknown parameters in the IGGM. Then the optimal parameter values chosen in advance were used to construct the final bathymetric model of the Philippine Sea. The four selected areas were as follows: A (21° N–27° N, 130° E–137° E), B (24° N–28° N, 138° E–145° E), C (8° N–15° N, 128° E–138° E) and D (10° N–16° N, 140° E–148° E), as shown in Fig. 4a. Area A is situated in the Daito Basin and the northern part of the Philippine Basin. Area B includes parts of the Shikoku Sea Basin, the Izu-Ogasawara Island Arc and Trench, along with other adjacent sea areas. Area C is mainly situated in the Philippine Basin and the Parece Vela Basin, with less topographic relief except for the Palau Ridge at the junction of the two basins. Area D is similar to Area B and has complex terrain, including the Parece Vela Basin, and the Mariana Trench with depths exceeding 10,000 m. Table 1 provides the removal statistics of the four areas, and Fig. 4b illustrates the distribution of shipborne bathymetric data after gross error removing.

**Table 1.** Removal statistics of shipborne bathymetric data according to the " $3\sigma$ " criterion with the GEBCO model serving as a reference.

| | Area | All points | Qualified points | Removing points | Removal rate |
|---|---|---|---|---|---|
| A | 130° E–137° E, 21° N–27° N | 57824 | 57237 | 587 | 1.02% |
| B | 138° E–145° E, 24° N–28° N | 79919 | 78424 | 1495 | 1.87% |
| C | 128° E–138° E, 8° N–15° N | 98267 | 96612 | 1655 | 1.68% |
| D | 140° E–148° E, 10° N–16° N | 405317 | 401120 | 4197 | 1.04% |



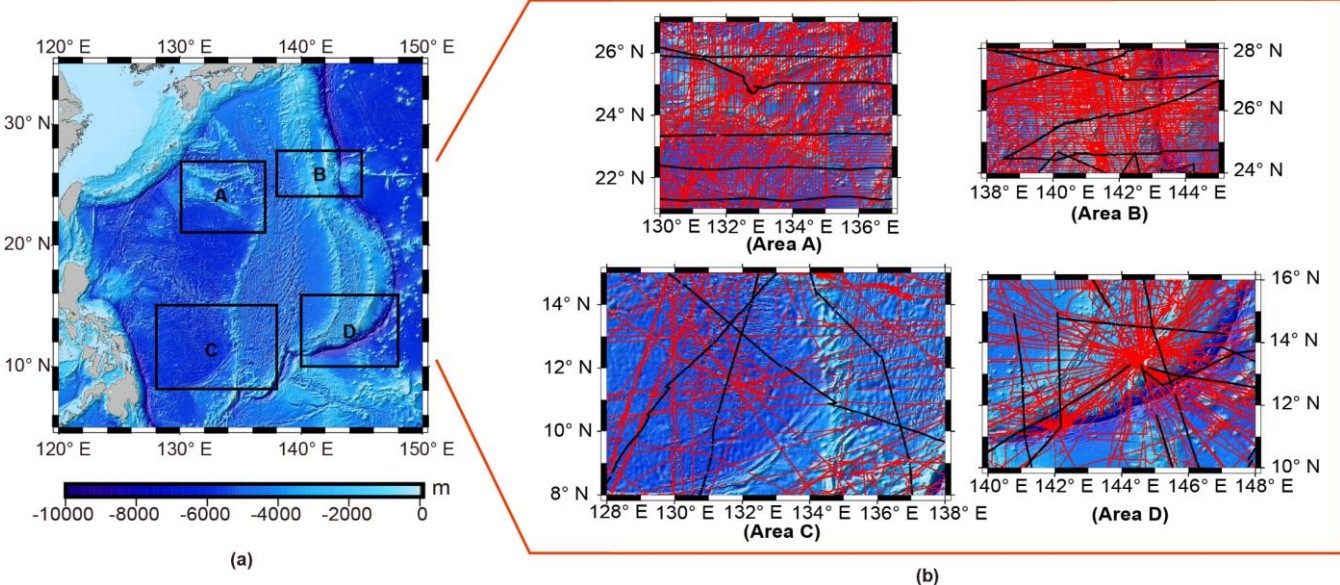

**Figure 4.** Distribution of four local areas and shipborne bathymetric data (red points are control points, black points are checkpoints, and the basemap is the GEBCO model).

To verify the applicability of IGGM in the Philippine Sea, corresponding bathymetric models were constructed by using

GGM and IGGM in four areas. Multiple independent cruises were selected within each area, which were not involved in the model calculations, for independent verification purposes (represented by the black points in Fig. 4b) to ensure the reliability of accuracy evaluation between GGM and IGGM. In both GGM and IGGM, the control points were divided into two parts. The first part was utilized for model calculation under different parameter values, while the second part was used for iteratively selecting the local optimal solutions of required parameters. Once the optimal values of each parameter were

determined through iteration, the final bathymetric model was calculated using all the control points. This two-step process allowed for a comprehensive evaluation of the parameter values and ensured that the final bathymetric model was calculated by the best possible combination of shipborne bathymetric data, gravity anomalies, and optimized parameter values. For the division of control points in the iterative method, the conventional GGM still adopted the original proportional selection method, where one control point for iteration was selected at an interval of three points on each cruise, as shown in Fig. 5.

On the other hand, IGGM divided control points using the cruise selection method based on the distribution of shipborne cruises, as shown in Fig. 6. The proportional selection method tended to consider the entire area for obtaining the optimal values of the parameter. Contrastingly, the cruise distribution-based selection weakened the effects caused by the interpolation of neighboring points.





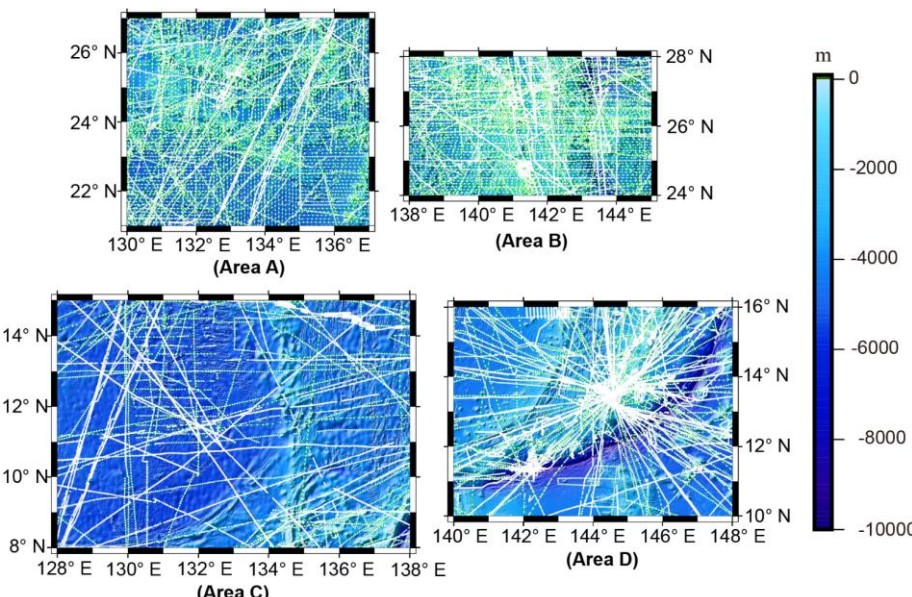

**Figure 5.** Proportional selection in GGM (The green control points were used to calculate bathymetric models corresponding to different unknown parameter values, and the white control points were employed for iteratively selecting the optimal value of $\Delta\rho$. The ratio of green dots to white dots was 3:1).

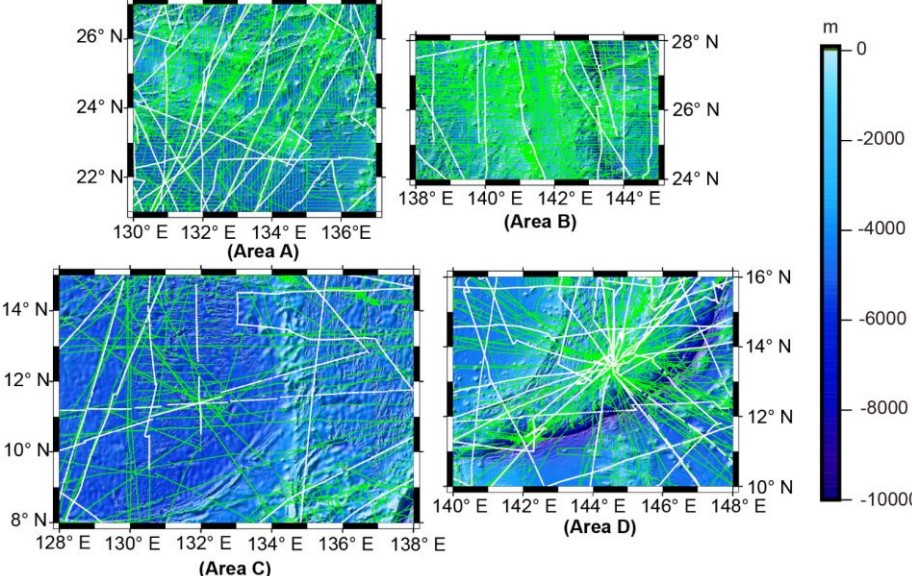

**Figure 6.** Selection based on the distribution of shipborne cruises in IGGM (The green control points are used to calculate bathymetric models corresponding to different unknown parameter values, and the white control points were employed for iteratively selecting the optimal parameter values of $\Delta\rho$, $R$ and $k$).

This section mainly involves the following steps. Firstly, the bathymetric models corresponding to the initial values of each parameter were calculated using green control points and were interpolated to obtain the predicted depth at white control





points. Secondly, the correlation coefficient and STD of predicted depths and shipborne measured depths at white control

points were calculated. The parameter values that yield the largest correlation coefficient and smallest STD (i.e. $\max_{1 \leq i \leq n} CC_i$ and $\min_{1 \leq i \leq n} STD_i$) were considered optimal. Subsequently, the final bathymetric model was constructed by combining all control points and using the optimal parameter values. The predicted depths at checkpoints were then obtained through cubic spline interpolation, and the prediction accuracies for both GGM and IGGM were evaluated based on the shipborne bathymetric data. The optimal parameter values for GGM and IGGM in the four regions were presented in Table 2.

It reveals that the optimal computational radius $R$ of IGGM is 2′ in different areas, but there are differences in the optimal values of other $\Delta\rho$ and $k$. Additionally, the density contrast value in Area D shows a significant difference, and the density contrast $\Delta\rho$ calculated by IGGM is closer to the theoretical value (1.67 g/cm³). Figure 7 displays the total number of surrounding points within a 2′ radius centered on each shipborne point, with black points indicating no other single-beam bathymetry point distribution. Statistical analysis shows that the numbers of black points in the four areas are 0, 454, 131,

and 130, respectively. And the percentages of surrounding shipborne points exceeding 20 are 72.1%, 59.1%, 72.9%, and 92.8% for the four areas, respectively. The optimal selection of density contrast in both GGM and IGGM enables the gravity anomalies to better characterize the distribution of seafloor topography (Nagarajan, 1994; Hu et al., 2012; Kim and Yun, 2018). However, it should be noted that the density contrast is used as an empirical parameter only for obtaining the optimal accuracy of the bathymetry model, thus diminishing its original physical significance.

**Table 2.** Optimal values of parameters for GGM and IGGM.

| Area | GGM | IGGM | | |
|---|---|---|---|---|
| | $\Delta\rho$ /(g/cm³) | $\Delta\rho$ /(g/cm³) | $R$ | $k$ |
| A | 0.6 | 0.6 | 2′ | 2.6 |
| B | 0.8 | 1.4 | 2′ | 6.4 |
| C | 1.0 | 0.9 | 2′ | 0.5 |
| D | 3.3 | 0.9 | 2′ | 3.8 |



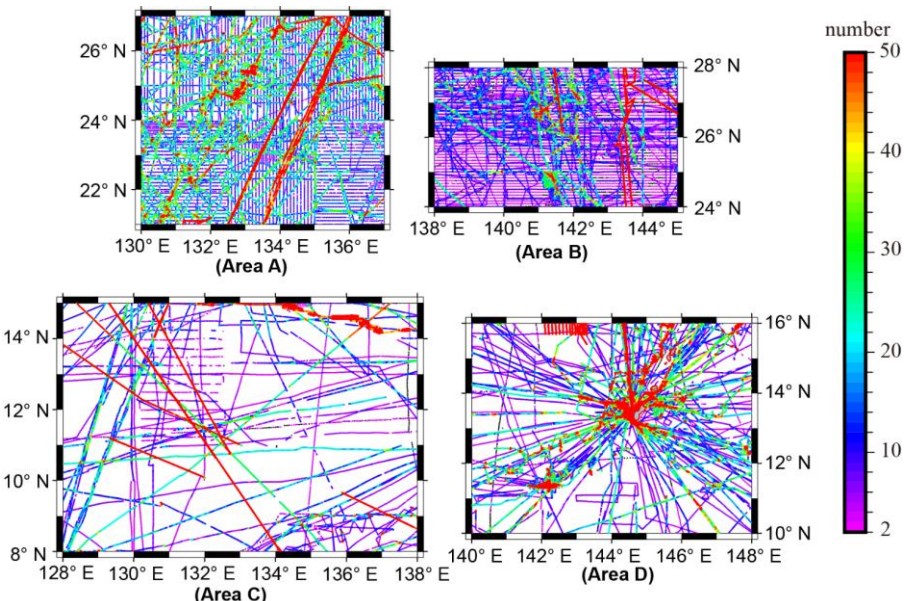

**Figure 7.** Total number of surrounding points within 2′ of each shipborne point as the center (black points represent no other single-beam shipborne points within a 2′ radius)

The corresponding bathymetric models constructed by GGM and IGGM were interpolated to obtain the predicted depths at the checkpoints. Table 3 displays the accuracy comparison between the predicted depth and the shipborne bathymetry in each area. Results indicate that the IGGM models in the four areas show varying degrees of accuracy improvement. The most significant enhancement is observed in Areas A and D, where the STD is reduced by approximately 16.36% (25.88 m) and 11.05% (33.39 m). In Areas B and C, the improvements of IGGM are limited, within 10 m, only slightly better than GGM models. The shipborne bathymetric data are evenly distributed in Areas A and B, but Area A is located in the Daito Basin with relatively slow topographic changes and exhibits the most substantial accuracy improvement. In contrast, there are Shikoku Basin and Izu-Ogasawara Island arc in Area B, whose eastern side is located at the junction of the Philippine Sea plate and the Pacific plate, and the topographic drop can reach up to thousands of meters. Terrain fluctuation is the main reason that affects the inversion accuracy, so the complex terrain in Area B leads to limited accuracy improvement of IGGM. Area C, positioned in the West Philippine Basin and the Parece Vela Basin, has relatively gentle terrain. However, the scarcity of sounding bathymetric data in this region, results in a large error when predicting checkpoint depths using gridding and interpolation processing. Although Area D includes the Mariana Trench with drastic topographic variations, it has a large number of shipborne bathymetric data.

The mean absolute percentage error (MAPE) is regarded as an indicator of relative accuracy, defined as the average value of the ratio (positive) of the prediction error to the measured depth:

$$MAPE = \frac{1}{n}\sum_{i=1}^{n}\left|\frac{E_i - H_i}{H_i}\right| \times 100\% , \tag{8}$$



where $H_i$ is the shipborne measured depth; and $E_i$ represents the predicted depth. A smaller MAPE indicates a higher
accuracy of the bathymetric model. The result shows that IGGM models in the four regions have higher relative
accuracy and can better characterize the geoinformation of seafloor topography. In conclusion, compared to GGM
models, IGGM models effectively improve accuracy, with the degree of improvement affected by the distribution of
250 shipborne bathymetry and terrain fluctuation. The significant improvement is observed in areas with flat terrain or
sufficient and uniform distribution of bathymetric data. Conversely, less data and larger terrain fluctuation result in
reduced accuracy improvement. Additionally, gridding and interpolation also introduce varying degrees of errors.

**Table 3.** Statistics of GGM and IGGM models at checkpoints (unit: m).

| Area | Model | Max | Min | Mean | STD | RMS | MAE | MAPE |
|------|-------|-----|-----|------|-----|-----|-----|------|
| | GGM-NCEI | 812.83 | -1174.47 | -37.31 | 158.19 | 162.49 | 93.49 | 2.71% |
| A | IGGM-NCEI | 542.65 | -953.26 | -23.85 | 132.31 | 134.42 | 78.74 | 2.40% |
| | GGM-IGGM | 700.28 | -366.72 | -13.46 | 81.06 | 82.15 | 47.69 | - |
| | GGM-NCEI | 1452.60 | -957.05 | -6.42 | 231.04 | 231.08 | 143.38 | 6.53% |
| B | IGGM-NCEI | 1417.82 | -931.33 | -1.77 | 223.39 | 223.39 | 139.37 | 6.35% |
| | GGM-IGGM | 392.61 | -502.48 | -4.65 | 53.25 | 53.44 | 27.82 | - |
| | GGM-NCEI | 907.11 | -1322.98 | -89.78 | 227.97 | 244.93 | 167.75 | 3.72% |
| C | IGGM-NCEI | 893.62 | -1315.04 | -84.26 | 219.31 | 234.87 | 160.64 | 3.58% |
| | GGM-IGGM | 372.07 | 352.01 | -5.52 | 61.85 | 62.07 | 39.51 | - |
| | GGM-NCEI | 1840.99 | -2876.56 | -8.56 | 302.15 | 302.21 | 182.19 | 5.19% |
| D | IGGM-NCEI | 1527.84 | -1991.20 | -8.51 | 268.76 | 268.84 | 174.17 | 4.81% |
| | GGM-IGGM | 794.94 | -1007.02 | -0.05 | 124.99 | 124.99 | 75.66 | - |

In Areas A, B and C, GGM model and IGGM models show relatively consistent bathymetry performance overall, as shown
in Fig. 8. However, notable differences are observed in Area D, particularly concentrated in the vicinity of the Mariana
Trench and trough, such as (10° N–12° N, 143° E–145° E), (12° N–13° N, 140° E–141° E), (10° N–11° N, 140° E–141° E)
and (15° N–16° N, 143° E–144° E). The GEBCO model, which incorporates the latest shipborne bathymetric data with high
accuracy, was used as the standard to evaluate the accuracy levels of GGM and IGGM models. Fig. 9a shows the distribution
of seafloor topography in the GEBCO model. Fig. 9b and Fig. 9c illustrate the differences between this model and the GGM
model, and IGGM model, respectively. Since the GEBCO model has a grid interval of 15 arcsec, it needs to be interpolated
to the corresponding grid points using the cubic spline. It is evident that the IGGM model exhibits smaller errors than the
GGM model and is closer to the GEBCO model within four ranges marked by black boxes in Fig. 9a. Especially, in the
vicinity of the Mariana Trench marked in the black box of Fig. 9a, the GGM model shows significant errors, further
indicating the higher precision performance of the IGGM model in Area D as depicted in Fig. 8. Within the red box range of
265 Fig. 9c, the IGGM model exhibits larger errors than the GGM model. This problem may be attributed to the lack of
shipborne bathymetric data in this range, resulting in the optimal parameters selected by IGGM being less applicable to a
certain local area. According to statistics, differences between the GEBCO model and the GGM and IGGM models within
the range of 0-300 m account for 86.2% and 88.8%, while the proportions of the differences above 500 m are 6.1% and 3.5%,
respectively. The GEBCO model, despite having a grid interval of 15 arcsec, exhibits lower true spatial resolution in the sea
area lacking of measured bathymetric data, as illustrated in the basemap of Area C in Fig. 6.







**Figure 8.** Comparison of the GGM model and IGGM model in four test seas.

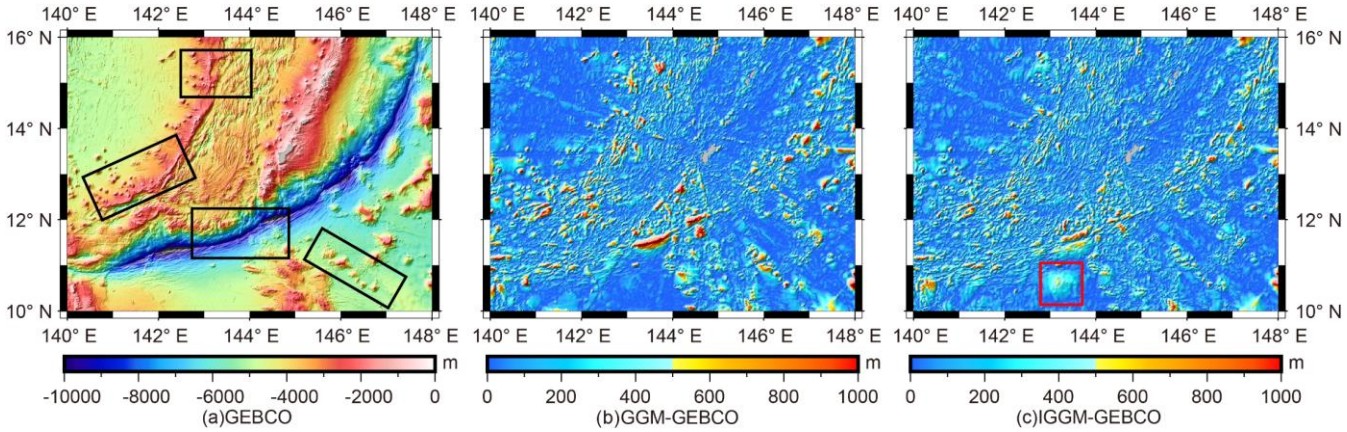

**Figure 9.** The GEBCO model and its comparison with the GGM model and IGGM model.

Based on the above statistics, it is evident that the IGGM, which considers the effect of regional seafloor topography, demonstrates higher accuracy at checkpoints, and achieves a significant improvement in the Philippine Sea. These results also confirm that the IGGM has better applicability. For all parameters required in IGGM, the optimal calculation radius in the four areas remains consistent at 2′, which is not affected by the distribution of seafloor topography and shipborne bathymetric data, but optimal values of $\Delta\rho$ and the index $k$ in the weight factor differ in each area. Given these findings, a predetermined calculation radius of 2′ is adopted for the final bathymetry modeling of the Philippine Sea (5° N−35° N, 120° E−150° E).

## 4.2 Bathymetric model based on IGGM in the Philippine Sea

Considering the huge amount of shipborne bathymetric data in the entire Philippine Sea and the limitations in computer capacity, the region (5° N–35° N, 120° E–150° E) is divided into nine subregions of 10°×10° each, as shown in Fig. 10a. To ensure data quality, the shipborne bathymetric data within each subregion are preprocessed by removing gross errors that exceed "$3\sigma$", with the GEBCO model used as a reference for comparison. Following the procedure of IGGM, the bathymetric models of each subregion are inverted. Finally, the 1′×1′ bathymetric model of the Philippine Sea (BAT_PS) is finally obtained, as depicted in Fig. 10b. During the construction of the BAT_PS model, the optimal parameters values required in IGGM are given in Table 4.



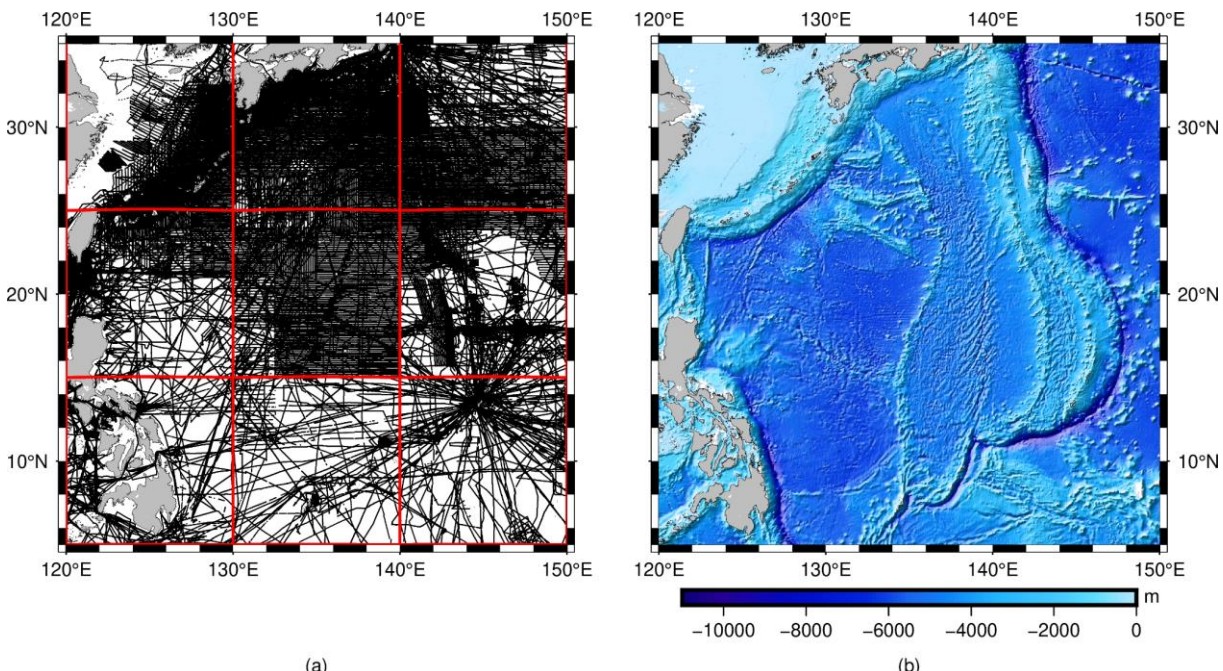

**Figure 10.** (a) Subregions of the Philippine Sea and distribution of shipborne bathymetric data. (b) BAT_PS bathymetric model constructed by IGGM in the Philippine Sea.

**Table 4.** Optimal values of $\Delta\rho$ and $k$ in each subregion during the construction of the BAT_PS model.

| | $\Delta\rho$ /(g/cm³) | $k$ | $\Delta\rho$ /(g/cm³) | $k$ | $\Delta\rho$ /(g/cm³) | $k$ |
|---|---|---|---|---|---|---|
| | 120° E–130° E | | 130° E–140° E | | 140° E–150° E | |
| 25° N–35° N | 1.0 | 1.7 | 1.0 | 3.9 | 0.7 | 1.5 |
| 15° N–25° N | 1.2 | 9.3 | 0.6 | 7.4 | 1.1 | 6.4 |
| 5° N–15° N | 2.0 | 2.5 | 1.2 | 6.2 | 1.0 | 6.4 |

The accuracy of the BAT_PS model was evaluated using three reference models: GEBCO, topo_25.1, and DTU18. The BAT_PS, GEBCO, topo_25.1, and DTU18 models were interpolated to all shipborne bathymetry points within the study area. As all the single-beam bathymetric data were involved in the model calculation, they could serve as an evaluation of internal consistency accuracy. The statistical results were presented in Table 5, with two categories of assessment: primary check (PC) and second check (SC). The primary check is the analysis of the difference between the predicted depth and the measured depth at all shipborne measured points. In the second check, the points with poor prediction quality based on the " $3\sigma$ " criterion are removed before conducting the statistical analysis. The results demonstrate that the BAT_PS model exhibits better agreement with the shipborne bathymetric data in the primary check, with STD and MAPE values of 89.81 m and 9.25%, outperforming the other three models. MAE represents the absolute magnitude of prediction errors, while MAPE reflects the percentage magnitude of prediction errors relative to the shipborne bathymetry. In the second check after removing gross errors, the accuracy levels of the BAT_PS model and the GEBCO_2022 model are approximately equal,




surpassing topo_25.1 model and DTU18 model. Overall, the accuracy ranking from high to low is BAT_PS model, GEBCO model, topo_25.1 model, and DTU18 model.

**Table 5.** Statistics of BAT_PS, GEBCO, topo_25.1, and DTU18 models at all shipborne bathymetric points (unit: m).

| Model | Max | Min | Mean | STD | RMS | MAE | MAPE | Removal rate | Remark |
|---|---|---|---|---|---|---|---|---|---|
| BAT_PS | 2509.61 | -2309.12 | 3.06 | 89.81 | 89.86 | 44.11 | 9.25% | - | PC |
|  | 269.42 | -269.42 | 2.03 | 59.58 | 59.62 | 36.63 | 8.94% | 1.86% | SC |
| GEBCO | 2334.08 | -2336.03 | -4.71 | 106.01 | 106.12 | 44.57 | 12.14% | - | PC |
|  | 318.03 | -318.03 | -3.54 | 60.10 | 60.20 | 34.95 | 11.56% | 1.79% | SC |
| topo_25.1 | 2490.43 | 2519.18 | 13.17 | 113.11 | 113.87 | 52.36 | 30.03% | - | PC |
|  | 339.28 | -339.32 | -11.34 | 69.62 | 70.54 | 42.58 | 27.48% | 1.79% | SC |
| DTU18 | 2460.07 | -3161.46 | 7.35 | 153.12 | 153.30 | 77.38 | 20.07% | - | PC |
|  | 459.32 | -459.35 | 2.82 | 105.16 | 105.20 | 63.28 | 19.43% | 2.22% | SC |

In the statistics of two check accuracies, both before and after gross error elimination, the DTU18 model outperforms the topo_25.1 model with MAPE values of 20.07% and 19.43%, compared to the topo_25.1 model's values of 30.03% and
27.48%. Although the topo_25.1 model has removed poor quality points during the second check, and other indicators, except MAPE, are also far better than those of the DTU18 model, the overall MAPE still remains worse than the DTU18 model. The topo_25.1 model may exhibit a large relative error in shallow seas. Figure 11 shows the relationship between the relative errors of four models and different depths. The relative error is described as the percentage of the prediction error and measured depth. MAPE can be regarded as a comprehensive statistical measure of relative error. As the depth increases,
the relative errors of each model exhibit a decreasing trend.

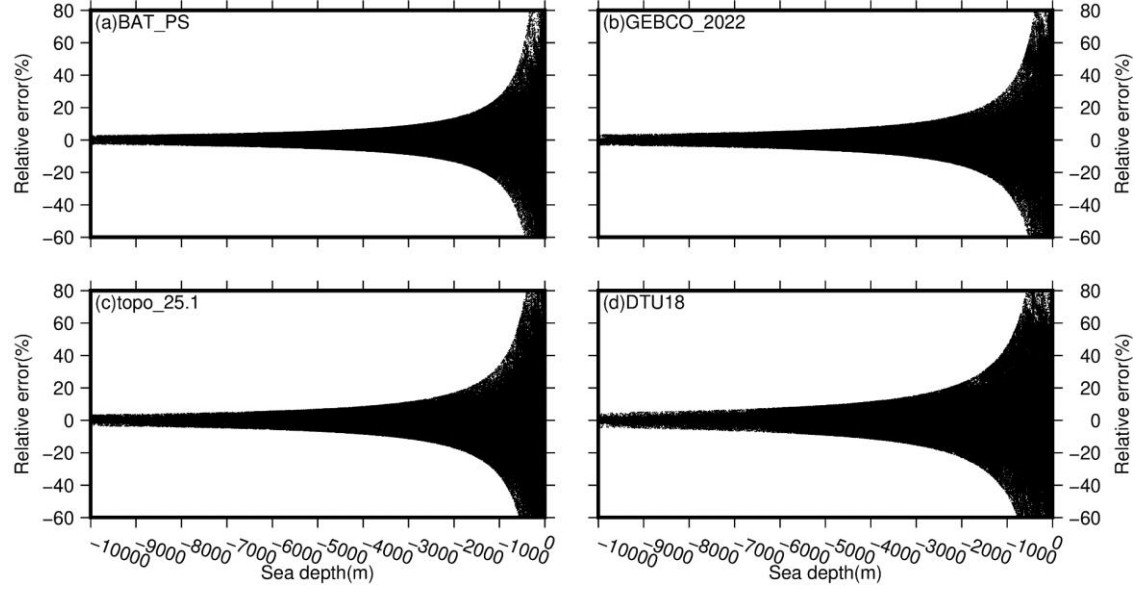

**Figure 11.** Relationship between relative errors and sea depths.

Table 6 further presents the statistics of MAPE at different depths after the second check. Specifically, the topo_25.1 model has a large relative error within a depth of 2000 m, which explains the phenomenon discussed above. After reaching a depth





greater than 2000 m, the relative error of the topo_25.1 model gradually becomes smaller than that of the DTU18 model. The construction of the BAT_PS model only relies on single-beam data without the inclusion of multi-beam bathymetric data. Consequently, the accuracy in some deep-sea areas is slightly lower. Additionally, the relative errors of each model increase when the depth exceeds -8,000 m. This can be attributed to most of these areas being near trenches, where both the accuracy of shipborne bathymetry and the precision of inversion methods face challenges due to the abrupt terrain.

**Table 6.** MAPEs for four models at different depths after the second check.

| Model<br>Depth(m) | BAT_PS | GEBCO | topo_25.1 | DTU18 |
|---|---|---|---|---|
| -2000 – 0 | 20.36% | 27.02% | 65.41% | 46.06% |
| -4000 – -2000 | 1.51% | 1.34% | 1.55% | 2.31% |
| -6000 – -4000 | 0.81% | 0.73% | 0.80% | 1.14% |
| -8000 – -6000 | 0.81% | 0.78% | 0.69% | 0.95% |
| -10000 – -8000 | 0.97% | 1.00% | 0.87% | 1.08% |
| -10000< | 1.10% | 1.09% | 1.49% | 1.49% |

## 5 Conclusions

Topography, especially bathymetric topography, has been a significant research focus with unique applications in the field of geoscience, although multiple techniques for bathymetric inversion and their corresponding improvement methods are temporarily constructed, a fully rigorous bathymetric inversion theory is yet to be established. Consequently, it is necessary 330 to refine and perfect the original method based on the current theory, serving as the foundation and focus of future bathymetric inversion research. In this paper, the IGGM, which considers the short-wavelength component effect of regional seafloor topography, is used to predict bathymetry in the Philippine Sea. The primary conclusions are summarized as follows:

(1) Four local areas in the Philippine Sea were selected as research subjects and corresponding bathymetric models were calculated using both GGM and IGGM. Overall, the accuracy of IGGM at checkpoints outperformed that of GGM, with 335 improvements of 16.36%, 3.31%, 3.80%, and 11.05%. This verifies the applicability of IGGM in the Philippine Sea. As a result, the optimal calculation radius of IGGM was set at 2' for the subsequent construction of the overall bathymetric model of the Philippine Sea.

(2) A high-precision BAT_PS model, with a grid of 1 arcmin, was established for the Philippine Sea (5° N–35° N, 120° E–150° E) using IGGM. The STD error between the predicted depths of the BAT_PS model and shipborne measured depths 340 reached 89.86 m at the single-beam shipborne points, and the accuracy of the BAT_PS model was 59.58 m without poor quality points. This accuracy is essentially equivalent to that of the GEBCO_2022 model and is significantly better than the topo_25.1 and DTU18 models.

(3) In this study, only single-beam bathymetric data were used for the construction of the BAT_PS model, while multi-beam bathymetric data are abundant in certain areas, such as the West Philippine Sea Basin, the Mariana Trench and Trough, were 345 not utilized. Future research will explore integrating multi-beam data into the model construction to further enhance the

accuracy of the BAT_PS model. Additionally, the accuracy of seafloor terrain inversion heavily relies on the ocean gravity field. The observation data from the Surface Water and Ocean Topography (SWOT) swath altimetry satellite are hoped to further enhance the resolution and accuracy of the marine gravity field, thereby greatly improving the prediction precision of bathymetry.

**Code availability**

All data used in this study are publicly available through the NCEI (https://www.ncei.noaa.gov/maps/bathymetry/), GEBCO (https://www.gebco.net/data_and_products/gridded_bathymetry_data/), SIO (https://topex.ucsd.edu/pub/global_grav_1min/), and DTU (https://ftp.space.dtu.dk/pub/DTU18/1_MIN/). The BAT_PS bathymetric model and source codes are available from this repository: https://doi.org/10.5281/zenodo.8351399.

**Author contributions**

Conceptualization: DA, JG. Methodology: DA, JG. Validation: DA, JG, XC, ZW, YJ, XL, VB, HS. Writing – original draft: DA. Writing – review & editing: DA, JG, XC, ZW, YJ, XL, VB, HS. All authors contributed to writing and revising the manuscript.

**Competing interests**

The contact author has declared that none of the authors has any competing interests.

**Acknowledgments**

We express our gratitude to the following organizations: NCEI for providing shipborne bathymetric data, GEBCO for offering bathymetric model, SIO for sharing gravity anomaly model and bathymetric model, and the DTU for providing bathymetric model. We also extend our appreciation to the editors and reviewers for their invaluable comments.

**Financial support**

This research has been supported by the National Natural Science Foundation of China (grant nos. 42192535, 42274006, and 42242015); and National Key Research and Development Program of China (grant no YS2018YFE011371).



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
