# Peer review of "High-precision 1'×1' bathymetric model of Philippine Sea inverted from marine gravity anomalies"

_EGUsphere, 2023_

## Referee Comment (RC2)

**Review of egusphere-2023-2132, "High-precision 1'×1' bathymetric model of Philippine Sea inversed from marine gravity anomalies"**

**General comments**

It is a good work; however, I think it is a repetition of the authors' work in the South China Sea (An et al. 2022). The IGGM was introduced for the first time in that paper, and was tested in the South China Sea. So, basically the regional capability of IGGM has already been proved in that work. With such considerable number of contributing authors, this study could (or should) have been a global test of the IGGM by merging a suite of regionally predicted depths. If not, then the only difference between this paper and the authors' previous paper is just the change in study area. And since each study area will yield different results, then any subsequent regional application of IGGM will just be a repetition.

Authors should check the units of measurements used. Some readers will argue that g/cm3 is not the generally accepted S.I. unit of density. Similar statement can be said of the unit of the gravitational constant.

**Specific comments**

Line 52: The S&S method is also a frequency domain method

Line 87: What do you mean by 'certain parameters?

Figure 1: The color palette used causes the empty grid cells (continent) to blend in as part of the gravity anomaly. Anomalies close to or greater than 80 mGal are showing as white, which is not far from gray. So, I suggest you consider changing this color palette.

You did not define  $\theta$  and k in Eqs. (3) and (4). How do you compute the value of  $\theta$ ? You should state it in the manuscript.

Check Eq. (5) again. The output of Eq. (4) is  $\Delta g_{res}^{j_n}$ , and not  $\Delta g_{short}^{j_n}$ .

Line 171: Change 'objects, analysing and determining' to 'subregions for analysing and determining'.

Line 171 - 172: The optimal parameters are chosen after determining the unknown parameters; am I right? If yes, then change 'in advance' to 'afterwards'.

Table 1: Change the column title 'Removing points' to 'No. of removed points'.

Figure 4: The geographic extents of Area A ( $6^{\circ} \times 7^{\circ}$ ) and Area D ( $6^{\circ} \times 8^{\circ}$ ) are almost similar; so, I think you could have plotted Area A to have slightly similar size as Area D. Apart from Area C, Areas A, B and D contain more depth data, but because you have plotted them too small in size, they look quite clumsy. The same applies to the subsequent related figures. Since you used GMT for the plots, you can even increase the plot size of Area B too. Use a different projection (-J option) for the *plot* module for each map.

Line 216 – 217: "and the density contrast  $\Delta \rho$  calculated by IGGM is closer to the theoretical value (1.67 g/cm3)". In reference to Table 2, this statement is very misleading. Because, apart from IGGM's 1.4 g/cm3 in Area B, the other values are far from 1.67 g/cm3, especially if you express the unit of density as kg/m3.

Figure 7: It would be better to change the color palette; it has made the figure kind of too flashy. I can hardly see the black points you are talking about, simply because of use of wrong color palette.

Moreover, there has been research commenting on the weaknesses of the rainbow color palette. You can look it up at https://doi.org/10.1016/j.isprsjprs.2022.10.002. Also, increase the sizes of the plots like I have commented in the preceding comment, so that they can be more readable.

Line 238: Change 'reason' to 'factor'.

Line 269 – 270: Change 'the sea area lacking of' to 'marine regions that lack'

You need to rephrase the first sentence of Line 310 - 312; it is not clear to understand.

Line 399: Check the spelling of the first author's name.

---

## Author Comment (AC1)

**Review response for *High-precision 1'×1' bathymetric model of Philippine Sea inversed from marine gravity anomalies**

Dear Reviewer,

Thanks for your useful suggestions and comments. We have carefully revised the manuscript according to the comments. Your opinions are reasonable, greatly helping me improve my article. The response to the reviewer's comments is as follows:

**Specific Comments**

1. Line30: why are there two trench-like terrains at (140°E-150°E, 30°N-35°N) and (125°E-127°E, 27°N-30°N)? This is clearly not true.

Response: We completely agree with this point and thanks for your careful observation. After investigation, we found that there were gross differences in the original shipborne bathymetric data on the two ship tracks (as showed in Extended Data Figure 1 and Figure 2). This part of the error was not effectively eliminated in the data preparation stage, which led to this situation. We have processed the bathymetric data again and recalculated the optimal bathymetric model (as showed in Extended Data Figure 3).

[Figure]

**Extended Data Figure 1.** IGGM model and shipborne bathymetric before correction in the region of 143°E-150°E, 32°N-34°N.

[Figure]

**Extended Data Figure 2.** IGGM model and shipborne bathymetric before correction in the region of 122°E-126°E, 26°N-30°N (Areas or points with water depths above 300 m are shown in black).

**Bathymetric model of Philippine Sea ( BAT_PS model) inversed from marine gravity anomalies**

[Figure]

**Extended Data Figure 3.** Modified BAT_PS model

2. Line 43, please accurately list several GEBCO models.

Response: Thank you for the suggestion, we have added an observation with this point. "Additionally, multiple GEBCO models **(such as GEBCO_2019, 2020, 2022, etc)** have been released."

3. Line 126, in Eq. (2), how are parameters $\Delta\rho$ and $D$ obtained?

Response: Thanks for your comments. We explain this question by referring to the statement in the paper: "$D$ is a reference datum, which is generally the deepest depth of control points. The parameters $\Delta\rho$ are calculated by iterative method in IGGM. The correlation coefficient and STD between the predicted depth and the shipborne bathymetry at points $i$ are analyzed under the conditions of different parameter values. When the correlation coefficient is the largest and the STD error is the smallest between predicted depths and measured depths (i.e. $\max_{1\leq i\leq n} CC_i$ and $\min_{1\leq i\leq n} STD_i$), the corresponding parameters are the optimal value."

4. Line 139, in Eq. (3), what does the letter "k" represent here? What is the influence of this parameter on the accuracy of bathymetric model inversion?

5.

Response: As you have highlighted, the parameter $'k'$ plays a crucial role in weighing the short-wavelength influence of surrounding points on the control points. It's used in conjunction with $'cos\theta_m'$, which is associated with the distance between the two points, and in this study the value of $'cos\theta_m'$ is between 0 and 1. Therefore, a larger $'k'$ represents a smaller proportion of the short-wavelength gravity effects caused by the surrounding points.

6. The elimination rate of data in Table 1 exceeds 1%, which will have an impact on the results of submarine terrain inversion.

Response: Thank you for your valuable comments on our paper. We appreciate your concern regarding the elimination rate of data exceeding 1% and its potential impact on submarine terrain inversion results. We want to clarify that the elevated rejection rate is attributed to the extensive time span of data collection, including instances before the advent of GPS technology, leading to poor positioning and notable measurement errors (Smith, 1993). To ensure the reliability of our shipboard bathymetric data, we must employ an effective rejection step to eliminate inaccurate or biased data. While we acknowledge the elimination rate surpassing 1%, our thorough analysis indicates that this does not significantly affect the final seafloor topographic inversion results. We are aware of this concern and plan to delve deeper into the impact of varying bathymetric data volumes on inversion results in future research to comprehensively assess the robustness of our approach.

7. When calculating the short-wave gravity at any point $i$, the long-wave gravity is derived from the long-wave gravity field, calculated using the tension spline function, and subjected to cubic spline interpolation. This entire calculation process inevitably introduces calculation errors. Does the paper address any relevant methods to mitigate or handle these errors?

Response: Thank you for your valuable comments. In our calculations for short-wave gravity at any point $i$, the long-wavelength gravity is derived from a tension spline function using GMT (Generic Mapping Tools) and is subsequently subjected to cubic spline interpolation. We recognize that this entire computation process may introduce errors. We believe that calculation errors in densely sampled areas of shipborne depth data are acceptable, while sparsely sampled areas may introduce noticeable errors. As of now, this issue has not been well-resolved, and our paper does not explicitly propose specific methods to mitigate these errors. We acknowledge the importance of addressing this issue and plan to explore and implement improved measures in our further research and revisions. Our goal is to reduce the impact of computational errors to the greatest extent possible, and we appreciate your feedback as it guides our ongoing efforts to enhance the accuracy and reliability of our methodology.

8. Fig7: The image is low resolution and I can't see the distribution of the black pints.

Response: Thanks for your input, we have redrawn Figure 7 in higher resolution to ensure better visibility of the distribution of the black points.

[Figure]

**Figure 7.** Total number of surrounding points within 2′ of each shipborne point as the center (black points represent no other single-beam shipborne points within a 2′ radius)

9. Fig9 (b) and (c): Unlike Fig8(c), the comparison between the GEBCO model and the GGM/IGGM model should be described as the absolute value of the difference.

Response: Thank you for your suggestions, we have added the appropriate explanations. "The GEBCO model and the absolute value of its differences in comparison with the GGM/IGGM models."

10. For the BAT_PS model and other reference models, there are still very large errors at some ship measurement points (Table 5). Can shipborne bathymetric data be used to further improve the accuracy of the model? For example, the difference at the ship measurement point can be added to the BAT model as a correction.

Response: We appreciate your thorough review of our paper and your invaluable suggestions. Your recommendation to use shipborne bathymetric data for model improvement is highly insightful. To further enhance the accuracy of the BAT_PS model, we have adopted your suggestion and implemented the following steps (SRTM, 2019):

(1). Interpolate the BAT_PS model to obtain predicted depths at ship measurement points and calculate the difference from the actual measured depths.

(1). Supplement grid points located 5 minutes away from the ship measurement points with zero-depth differences. These additional zero-depth data points prevent the interpolation algorithm from generating abrupt changes in gaps adjacent to areas with rapid depth variations.

(3). Use the GMT module "surface" to generate a corrected grid (Figure 12(a)) by combining the depth differences at ship measurement points and the zero values at grid points.

(4). Restore the corrected grid to the BAT_PS model to obtain the CBAT_PS (constrained BAT_PS) model, as shown in Figure 12(b).

We also added the corresponding explanation in the paper and the final model has been uploaded to *Zenodo* (https://zenodo.org/records/8351399).

[Figure]

**Figure 12.** (a) Depth correction grid constrained by shipborne bathymetry. (b) CBAT_PS model constrained by shipborne bathymetry.

11. Is it reasonable to determine the computational radius to be 2′ for the entire Philippine Sea area? In general, re-selecting the optimal value for each sub-region is required to obtain a locally optimal model.

Response: Thank you for your valuable suggestions and comments. Regarding the predetermined computational radius of 2′ for the entire Philippine Sea area, our decision is based on two primary considerations. Firstly, numerous experiments have revealed that, within a certain range, variations in the calculation radius do not significantly impact model accuracy. Additionally, given the substantial amount of data available for the Philippines, we aim to improve calculation efficiency while ensuring accuracy. Predetermining this parameter in advance is a pragmatic approach. We are also in the process of applying this approach to global model building. We appreciate your thoughtful input, and we believe that these considerations contribute to a fair and effective modeling strategy.

12. Line 287: How are the boundary points of each region treated? How are the subregions stitched together?

Response: Thank you for your comment. The stitching of subregions is accomplished using GMT. For the coincidence points at the boundary, their average value is taken as the value of respective node.

**Technical corrections**

1. In line 47, the sentence "as compared to other observation methods" can be changed to "compared to alternative observation methods".

Response: We thank the reviewer for rising this point. We have revised this sentence to make it more understandable.

2. In line 59, the first occurrence of 'SIO' should be provided in full.

Response: We thank the reviewer for rising this point. We have revised this sentence to make it more understandable.

3. In lines 74-77, the sentence is too long to convey its meaning accurately.

Response: We thank the reviewer for rising this point. We have removed burdensome expressions to make it clearer.

4. In line 100, please provide an accurate explanation for '3b' to ensure understanding for all readers.

Response: Thank you for the suggestion, we have rewritten the sentence to improve understanding for all readers: "Interpolate the GEBCO_2022 model to the shipborne bathymetry points, calculate the difference with the shipborne depth, and then exclude points where the difference exceeds 3 times the standard deviation (STD) of the whole differences."

5. Line105: "is a continuous global terrain model".
Response: Noted.

6. Figure 3 needs to be redrawn, the text in the figure is not clear, and the flowchart logic is confused.

Response: We thank the reviewer for rising this point. Figure 3 has been redrawn to enhance clarity, making the text more legible and improving the logic of the flowchart.

[Figure]

**Figure 3.** Flowchart of the IGGM.

7. Line218: Does "the total number of surrounding points within a 2′ radius centered on each shipborne point" include the centre point? The minimum value of colorbar in Fig7 starts at 2.

Response: Yes, the representation in the previous manuscript included the central point. However, we have redrawn Figure 7 and its colorbar for clarity. In the updated version, 'the total number of surrounding points within a 2' radius' does not include the center point.

8. Check the formatting of references, e.g. Lines 394, 402, 429, 437, 460.

Response: Noted. Thanks for your comments.

9. Line 153: "value" should be "values".

Response: Noted.

10. Line345: "were not utilized" should be deleted.

Response: Noted.

---

## Author Comment (AC2)

**Review response for *High-precision 1'×1' bathymetric model of Philippine Sea inversed from marine gravity anomalies**

Dear Reviewer,

   Thanks for your useful suggestions and comments. Your opinions are significant, greatly helping me improve my article. We have carefully revised the manuscript according to the comments and hope that the correction will meet with approval. The response to the reviewer's comments is as follows:

**General comments**

*It is a good work; however, I think it is a repetition of the authors' work in the South China Sea (An et al. 2022). The IGGM was introduced for the first time in that paper, and was tested in the South China Sea. So, basically the regional capability of IGGM has already been proved in that work. With such considerable number of contributing authors, this study could (or should) have been a global test of the IGGM by merging a suite of regionally predicted depths. If not, then the only difference between this paper and the authors' previous paper is just the change in study area. And since each study area will yield different results, then any subsequent regional application of IGGM will just be a repetition.*

Response: Thank you for recognizing our research and providing valuable suggestions. We highly value your comments on our work and would like to use this response to express our research motivation and methodology more clearly. Our original intention in writing this paper was not to replicate our previous work, as you rightly pointed out, we aimed to apply the method globally. Before its global application, we conducted experiments in the representative Philippine Sea with the goal of optimizing the selection strategy for relevant parameters and releasing a reliable bathymetry model. Additionally, we need to verify the method's universality, not just its applicability to a specific region. Currently, uncertainties persist regarding the inversion accuracy on a global scale. We noted the importance and uniqueness of the Philippine Sea, a region that is clearly characterized by shipboard bathymetric data and the distribution of seafloor topography. Therefore, in this phase we focus on showing the potential global applicability of the method through experiments in the Philippine Sea. We would like to discover more interesting aspects, such as how the inversion strategy and result from the Philippine Sea can be utilized globally. All co-authors and contributors aspire to expand this work globally in the future, and we are diligently working towards that goal. Moreover, numerous new issues need addressing in the global seafloor topography modeling effort. Consequently, we are making our algorithmic program available to the public, hoping to receive suggestions and corrections from experts and scholars to enhance our research. Thank you again for your valuable comments, and we will fully consider your suggestions.

*Authors should check the units of measurements used. Some readers will argue that g/cm³ is not the generally accepted S.I. unit of density. Similar statement can be said of the unit of the gravitational constant.*

Response: Thanks for your suggestions, we have corrected the relevant units by replacing $g/cm^3$ with $kg/m^3$, and the gravitation constant $6.672×10^{-11} \ m^3/kg·s^2$.

**Specific comments**

*Line 52: The S&S method is also a frequency domain method*

Response: Noted. Thanks for your comments.

*Line 87: What do you mean by 'certain parameters?*

Response: Thanks for your comments. The expression here is slightly ambiguous, and 'certain parameters' represent the density constant $\Delta\rho$, the calculated radius, and the weight parameter $k$.

*Figure 1: The color palette used causes the empty grid cells (continent) to blend in as part of the gravity anomaly. Anomalies close to or greater than 80 mGal are showing as white, which is not far from gray. So, I suggest you consider changing this color palette.*

Response: Thanks for your comments. We have changed this color palette.

[Figure]

**Figure 1.** SIO V32.1 gravity anomaly model in the study area.

*You did not define θ and k in Eqs. (3) and (4). How do you compute the value of θ? You should state it in the manuscript.*

Response: Thank you for your suggestions, we have added the appropriate explanations. "where $\theta$ is calculated by the arctangent value based on the depth of the control point and the horizontal distance between the control point and the surrounding sea surface points. And $k$ is an unknown parameter determined by the iterative algorithm and is used in conjunction with $cos\theta_m$ as a weighting parameter to quantify short-wavelength gravity effects caused by the surrounding points."

*Check Eq. (5) again. The output of Eq. (4) is $\Delta g_{res}^{j_n}$, and not $\Delta g_{short}^{j_n}$.*

Response: Noted. Thank you for your comments and careful review.

*Line 171: Change 'objects, analysing and determining' to 'subregions for analysing and determining'.*

Response: Noted. Thank you for your comments.

*Line 171 – 172: The optimal parameters are chosen after determining the unknown parameters; am I right? If yes, then change 'in advance' to 'afterwards'.*

Response: Yes, you're right. We've changed it.

*Table 1: Change the column title 'Removing points' to 'No. of removed points'.*

Response: Noted. Thank you for your comments.

*Figure 4: The geographic extents of Area A (6º×7º) and Area D (6º×8º) are almost similar; so, I think you could have plotted Area A to have slightly similar size as Area D. Apart from Area C, Areas A, B and D contain more depth data, but because you have plotted them too small in size, they look quite clumsy. The same applies to the subsequent related figures. Since you used GMT for the plots, you can even increase the plot size of Area B too. Use a different projection (-J option) for the plot module for each map.*

Response: Thank you for your comments and suggestion. We completely agree with you and have redrawn the relevant figures to make them more appropriate.

[Figure]

**Figure 4.** Distribution of four local areas and shipborne bathymetric data (red points are control points, black points are checkpoints, and the basemap is the GEBCO model).

[Figure]

**Figure 6.** Selection based on the distribution of shipborne cruises in IGGM (The green control points are used to calculate bathymetric models corresponding to different unknown parameter values, and the white control points were employed for iteratively selecting the optimal parameter values of $\Delta\rho$, $R$ and $k$).

[Figure]

**Figure 7.** Total number of surrounding points within 2′ of each shipborne point as the center (black points represent no other single-beam shipborne points within a 2′ radius)

*Line 216 – 217: "and the density contrast $\Delta\rho$ calculated by IGGM is closer to the theoretical value (1.67 g/cm³)". In reference to Table 2, this statement is very misleading. Because, apart from IGGM's 1.4 g/cm³ in Area B, the other values are far from 1.67 g/cm³, especially if you express the unit of density as kg/m³.*

Response: Thank you for your comments. We completely agree with you and we have removed this statement.

*Figure 7: It would be better to change the color palette; it has made the figure kind of too flashy. I can hardly see the black points you are talking about, simply because of use of wrong color palette. Moreover, there has been research commenting on the weaknesses of the rainbow color palette. You can look it up at https://doi.org/10.1016/j.isprsjprs.2022.10.002. Also, increase the sizes of the plots like I have commented in the preceding comment, so that they can be more readable.*

Response: Thank you for your comments and suggestion. We have changed this color palette.

*Line 238: Change 'reason' to 'factor'.*

Response: Noted. Thank you for your comments.

*Line 269 – 270: Change 'the sea area lacking of' to 'marine regions that lack'*

Response: Noted. Thank you for your comments.

*You need to rephrase the first sentence of Line 310 – 312; it is not clear to understand.*

Response: Thank you for your comments. We have rephrased the sentence to make it clearer.

"In the second checks the topo_25.1 model has excluded poor-quality points, with both the STD and RMS showing better performance compared to the DTU18 model. However, it is noteworthy that the MAPE of topo_25.1 is worse."

*Line 399: Check the spelling of the first author's name.*

Response: Noted. Thank you for your comments and careful review.